# The Effects of Drought Stress Intensity and Duration on the Dynamics of Nonstructural Carbohydrates in *Pinus yunnanensis* Seedlings

**DOI:** 10.3390/plants14060980

**Published:** 2025-03-20

**Authors:** Xin Deng, Xin Chen, Tianyu Li, Hang Zhang, Yun Bao, Jingwen Yang, Li Zheng, Ping Lan, Junwen Wu

**Affiliations:** 1Yunnan Academy of Ecological and Environmental Sciences, Kunming 650034, China; 15501270919@163.com (X.D.); 18587198456@163.com (X.C.); litianyu9507@outlook.com (T.L.); ynzh8990@126.com (H.Z.); kmyunyun310@163.com (Y.B.); 13658893729@163.com (J.Y.); 2Yunnan Engineering Research Center of Heavy Metal Pollution Control, Kunming 650028, China; 3Yunnan Appraisal Center for Ecological and Environmental Engineering, Kunming 650028, China; 4College of Forestry, Southwest Forestry University, Kunming 650224, China

**Keywords:** drought stress intensity, drought duration, *Pinus yunnanensis* seedlings, non-structural carbohydrates

## Abstract

How drought impacts the allocation of nonstructural carbohydrates (NSCs) in *Pinus yunnanensis* remains unclear. In this study, *Pinus yunnanensis* seedlings were subjected to four levels of drought stresses treatment during a 60-day experiment period, including suitable moisture (CK), light drought (LD), moderate drought (MD), and severe drought (SD). NSCs in needles, stems, coarse roots, and fine roots were measured every two weeks. The distribution of NSC in *P. yunnanensis* seedlings varied with both drought stress intensity and duration, with different organ-specific patterns under increasing drought stress. Before the first 15 days, the intensity of drought stress had no significant effect on needle, stem and coarse root NSC contents, but decreased fine root NSC contents significantly. Between day 30 and 45, drought stress intensity showed no significant effect on NSC content in all organs. However, at 60 d, compared with CK, stem NSC concentrations under MD and SD increased by 47.92% and 48.23%, whereas fine root NSC concentrations decreased by 23.38% under SD conditions. With the extension of the drought duration, coarse root NSC increased while fine root NSC content decreased under SD conditions. Our results highlight the important role played by drought duration in controlling the NSC dynamics. Only fine root NSC decreased at the initial stage, and day 60 emerged as a turning point at which organ-level NSC changes became noticeable. These findings provide great insights into the understanding of organ-specific NSC dynamics under drought stress.

## 1. Introduction

Rapid changes in the global climate have increased the frequency and intensity of drought events, creating worrying phenomena such as forest decline [1]. In particular, rainfall patterns are changing across the globe, drought-induced forest mortality is increasing, plants in almost every forest biome are living at the edge of their survival hydrological limits, and forest ecosystem services have been severely impacted [1,2,3]. Extreme droughts inhibit growth and cause hydraulic damage in trees, which can lead to tree mortality, carbon starvation and a rise in forest degradation [3]. The Yunnan province in China, a region seriously affected by increasing winter–spring drought duration and intensity, has been experiencing increased drought-induced tree mortality in its forests in recent years [4]. The increasing role of drought in community dynamics and forest mortality has led to a growing interest in understanding how to enhance drought resistance [2].

Previous analyses of the death mechanism show that there are two prevailing hypotheses related to drought-induced tree death: carbon starvation and hydraulic failure, with the latter considered being more common [5]. Drought can directly cause trees to suffer from consequences such as embolism, hydraulic failures and cell failure and can also affect the carbon balance of trees [6]. When faced with long-term drought stress, plants experience a decrease in photosynthesis through leaf loss and stomata closure, resulting in increased consumption of stored carbon to meet their metabolic needs [5]. When carbon cannot maintain the basic functions of trees, “carbon starvation” occurs, which may eventually lead to the death of tree [5,7]. One of the defense mechanisms of plants against drought is their ability to adjust the allocation of nonstructural carbohydrates (NSCs) among different organs. NSCs—mainly consisting of varying compositions of soluble sugar and starch—characterize the carbon budget and balance in plants and affect their resistance to stress [8]. Under drought stress, plants regulate their NSC changes, which indirectly affect other physiological mechanisms [9]. Therefore, the dynamics of NSC play a key role in drought adaptation [10]. However, due to its complex and diverse physiological processes, no consistent trends have been observed thus far [11]. One of the key complexities is that the NSC response to drought can differ widely among species, ages, and drought intensities and durations [12].

Firstly, different drought intensities may induce different NSC responses of trees [11]. For instance, it is reported that no NSC consumption is observed under mild or moderate drought stress in *Pinus edulis* and *Juniperus osteosperma* [5]. O’Brien (2015) found that under moderate drought stress, NSC in woody tissue decreased, while it increased under extreme drought in shade-tolerant tropical tree seedlings [13]. In the case of conifer species, it was found that the hydraulic conductivity and carbohydrate contents decreased under extreme drought in *Pseudotsuga menziesii* (Mirb.) Franco, *Pinus ponderosa* Douglas ex C. Lawson, and *Pinus lambertiana* Douglas [14].

Secondly, the dynamics of NSC also vary with the duration of drought stress [5]. During the early stages of drought stress, growth tends to decline more rapidly than photosynthesis [15,16], which may subsequently induce an increase in NSC content in trees. This early-stage NSC increase is observed in *Quercus coccifera*, *Arbutus andrachne*, *Pistacia lentiscus* and *Olea europaea* [15]. As the drought stress increases, the growth rate of trees may slow down or even stop, and photosynthesis and respiration rate also decrease, which may eventually decrease the NSC concentrations in trees [5].

Moreover, different organs may show contrasting responses to drought. For example, root NSC is reported to decrease in *Ulmus minor* and *Quercus ilex* with increasing water stress, yet the leaf and stem NSC increase in *Quercus ilex*, while leaf NSC decreases in *Ulmus minor* [17,18]. Under severe drought conditions, NSC concentrations in *Picea abies* roots decrease significantly [12]. In a word, drought-induced changes in carbon allocation, utilization, and transport differ between above- and below-ground tree organs [19], during different stages of drought, and under different drought intensities.

*Pinus yunnanensis*, an endemic and a major timber species in southwestern China, has been used as a pioneer tree species for the reforestation of barren mountains, accounting for about 52% of the forest area in Yunnan Province [20]. Although many studies on drought stress and NSC dynamics have been conducted over the past 20 years, covering a range of species (e.g., *Laurus nobilis* [21], *Pinus massoniana* [22], *Pinus sylvestris*, and *Picea abies* [23]), little is known about how drought intensity and duration affect NSC in *P. yunnanensis*, leaving a significant knowledge gap in our understanding of drought resistance mechanisms under a changing climate. To address this gap, we hypothesize that (1) NSC content in *P. yunnanensis* decreases only in severe drought and is related to the duration of drought, and (2) NSC in *P. yunnanensis* does not change during the initial stages of drought stress, while it decreases in the later stages.

## 2. Results

### 2.1. Effect of Drought Stress on NSC

We conducted a two-way ANOVA on the data from different organs, with 4–6 replicates (Table 1)**.** Drought duration exhibited a significant effect on soluble sugar, starch, NSC content and soluble sugar-to-starch ratios in all organs. Drought intensity exerted significant effects on fine root soluble sugar, stem starch, stem and fine root NSC, and needle soluble sugar-to-starch ratios. Furthermore, a significant interaction effect was observed for drought stress duration and intensity on the coarse root NSC, needle soluble sugar-to-starch ratios.

### 2.2. Soluble Sugar Concentrations

Different seedling organs (needles, stems, coarse and fine roots) exhibited different responses to drought intensity and progression in terms of sugar concentration (Figure 1A–D). Soluble sugar content of needles, fine roots and coarse roots in each treatment showed an increasing trend until day 15 and then a decreasing trend afterwards. Differences between treatments occurred only at 60 days where moderate and severe drought exhibited higher values. The concentration of soluble sugar in needles and coarse roots under SD increased by 23.50% and 31.63%, respectively, whereas it decreased in fine roots by 24.61%.

### 2.3. Starch Concentrations

Although starch content in all organs first declined (up to day 30), and then increased at day 45, followed by a decrease at day 60, these changes were only significant in needles and stems (Figure 2A–D). Under LD treatment at day 60, needle starch content was 40.31% higher than CK. Under SD treatment at day 60, stem starch content was 119.26% higher than CK.

### 2.4. NSC Concentrations

There were no significant differences in needle and coarse roots under different drought stress treatments and across all time periods. However, stems at day 60, and fine roots at days 15 and 60 showed some significant differences. At 15 d, fine root NSC concentrations under LD were 31.11% lower than CK. At 60 d, compared with CK, stem NSC concentrations under MD and SD increased by 47.92% and 48.23%, whereas fine root NSC concentrations decreased by 23.38% under SD (Figure 3A–D).

### 2.5. Soluble Sugar-to-Starch Ratio

Sugars and starch are interconvertible, and the ratio of sugars to starch reflects the dynamic changes in their contents, which serves as an analytical indicator in related studies [22]. At 15 d, soluble sugar-to-starch ratios of needles under LD, MD and SD were significantly higher than those under CK. During the whole experiment, the ratio in each organ was greater than 1, and the ratio of MD and SD in the stem decreased at 45 d. On day 60 of drought stress, substantial changes and distinct differences in ratios were observed in the needles (Figure 4A–D).

### 2.6. NSC Distribution Pattern

During the first 30 days, NSC allocation proportions among organs changed only slightly, but larger shifts occurred from day 45 to day 60. At day 60, compared to day 15, the LD treatment was observed to increase NSC concentrations in the needles and stems by 8% and 3%, respectively. Meanwhile, coarse and fine roots were found to have NSC decreases of 10% and 1%, respectively (Figure 5A–L); the MD treatment increased needle and stem NSC by 3% and 14%, respectively, and decreased coarse and fine root NSC by 11% and 6%, respectively (Figure 5A–L); and the SD treatment decreased needle, coarse root and fine root NSC by 10%, 1% and 2%, and increased stem NSC by 12% (Figure 5A–L).

## 3. Materials and Methods

### 3.1. Experimental Site

This research was conducted at the Arboretum of Southwest Forestry University, Kunming, Yunnan province, China (102°46′ E, 25°03′ N). The site lies in a subtropical plateau monsoon climate zone. At an elevation of 1964 m, the climate here is mild with a short frost period, an average annual temperature of 16.5 °C, average annual precipitation of 1035 mm, and average annual relative humidity of 67%. The soil type is red loam. The temperature inside the shelter (4 m high, 10 m long, 3 m wide) ranges from 18.5 to 37.0 °C, with relative humidity between 22.3 and 48.0%. The top of the site is sheltered by a white plastic film that lets through sunlight from the top, but excludes the rain, and the sides are left open for airflow.

### 3.2. Seedling Preparation

Seeds of *P. yunnanensis* were obtained from a seed orchard in Midu County, Dali City, Yunnan Province. The seeds were cultivated in seedling bags and grown in nursery for 2 years. On 15 August 2020, 200 two-year-old *P. yunnanensis* seedlings were selected from Yiliang Garden Forestry, and transplanted into plastic flowerpots. The pots have a diameter of 26.5 cm, a base diameter of 21 cm, and a height of 21 cm, and each was filled with 6 kg of sieved soil mixture (red loam: humus ratio = 3:2), one seedling per pot, and a tray at the bottom. The transplanted seedlings were placed on the experimental plot, which was covered with a layer of mulch to prevent interference from underground moisture. A rain shelter was built to ensure good ventilation without shading. The soil water content of the potted plants was maintained at field holding capacity to ensure the healthy growth of the seedlings. After the seedlings were transplanted, the root collar diameter and height of all the seedlings were measured with vernier calipers and measuring tapes, with an accuracy of 0.01 mm and 0.1 cm, respectively. Initial values of seedling height and ground diameter were 20.84 ± 1.70 cm, 18.60 ± 0.47 mm, respectively.

### 3.3. Application of Drought Treatments

Seedlings with similar growth were selected, labeled and divided into four groups of 40 each. The height and ground diameter of all seedlings were measured and recorded during the experiment. At the start of the experiment, the field water holding capacity of the soil was 25.94% (determined by the ring knife method [22]), and soil bulk density was 1.14 g·cm^−3^. During the experiment, real-time measurements of soil volumetric moisture content were collected with a soil moisture meter (Decagon Inc., Washington, DC, USA).

Four moisture treatment methods were used to simulate different natural drought conditions, based on percentages of field capacity: (1) suitable moisture (CK, 80% ± 5%), (2) light drought stress (LD, 65% ± 5%), (3) moderate drought stress (MD, 50% ± 5%) and (4) severe drought stress (SD, 35% ± 5%). These correspond to actual moisture contents of 19.45–22.05%, 15.56–18.16%, 11.67–14.27%, and 7.78–10.38%, respectively. Actual soil water content was measured daily at 17:00, using a Procheck handheld multifunctional soil moisture meter (Decagon, USA). After all the potted plants were weighed at 17:00 and soil water content measured, the target moisture level was maintained by watering or withholding water to reach the designated weight. The experiment started on March 14 and ended on 12 May 2021, lasting 60 days in total.

### 3.4. Measurements and Sampling

Growth measurements and destructive sampling were taken on the 15th, 30th, 45th, and 60th days into the drought stress experiment (i.e., 29 March, 13 April, 28 April, and 12 May, respectively). The sampling time was fixed at 17:00 on the sampling date because the NSC concentration in the leaves fluctuates with the photosynthesis activity [24]. At each sampling, 4–6 seedlings were selected per treatment. Needles, stems, coarse roots (root diameter > 2 mm) and fine roots (root diameter < 2 mm) from these samples were washed separately with distilled water. To prevent enzymatic carbohydrate reactions, the seedlings were placed in an oven at 105 °C for 30 min, and then dried at 80 °C to a constant dry weight [22,25].

Dried samples were ground to pass through a 60-mesh screen. The total NSC was assessed by summing soluble sugars and starch contents, both determined using the phenol colorimetric method [22,25,26]. First, 0.300 g of dry sample was put into a 10 mL centrifuge tube with 4 mL of 80% ethanol, and heated at 80 °C for 30 min. Centrifuged and cooled at 3000 r/min for 10 min, the supernatant was poured into a graduated test tube, and the residue was added to 2 mL of 80% ethanol. We repeated the same extraction twice, and the soluble sugar content was determined by the standard curve of anthrone colorimetry at 625 nm [26]. Second, the precipitate after the extraction of soluble sugar was pasteurized with distilled water for 15 min, and then extracted with 2 mL of 9.2 mol/L perchloric acid for 15 min. The precipitate was centrifuged after cooling and the supernatant was collected. The sugar content in the extract was obtained colorimetrically at 625 nm, and multiplied by 0.9 to give the actual starch content (to account for the water in the starch).

### 3.5. Statistical Analysis

All statistical analyses were performed using SPSS 19.0 (IBM). Values are expressed as mean ± standard deviation, with 4–6 replicates per treatment. The data were tested for normality and homogeneity of variance prior to analysis of variance. Two-way factor analysis of variance (ANOVA) was used to analyze the effects of drought intensity (CK, LD, MD, and SD) and drought duration (15, 30, 45, and 60 days) on the NSC (sugar and starch) content in each organ. The differences in soluble sugars, starch, NSC content and biomass were examined by one-way ANOVA with different water treatment levels and different treatment times. Duncan’s method was used for multiple comparisons, at a 0.05 level of significance. Graphs were plotted using Origin (2021 Edition, Origin Lab Company, Miami, FL, USA). Biomass proportions for needles, stems, coarse roots, and fine roots were calculated relative to total biomass, while organ-level NSC allocation was obtained by multiplying organ biomass by organ-level NSC content and the total NSC storage was the sum of NSC reserves in all organs.

## 4. Discussion

### 4.1. Effects of Drought Intensity on NSC Dynamics

The effects of drought intensity on NSC concentrations may arise from changes in carbon sources, sinks and related regulatory mechanisms. Previous studies showed that in the presence of carbohydrate deficiency, trees may prioritize investment in NSC storage over growth [11,27]. Furthermore, Wiley found that NSC remained stable under moderate drought stress and significantly decreased under severe drought stress [28]. In our study, significant changes in NSC were only found in severe and moderate drought treatments in stems, and in severe drought treatment in fine roots. Furthermore, it happens mainly at the later stage of drought (day 60, Figure 3). This confirms the first hypothesis of this study that NSC content in *P. yunnanensis* decreases only in severe drought and is related to the duration of drought. After 60 days of moderate drought treatment, Deng observed that the root NSC content of 2-year-old *Pinus massoniana* seedlings increased [25]; this is contrast to the results of our study that more NSC accumulation occurs in stems and less NSC in fine roots. On one hand, severe drought may significantly limit tree growth and development, leading to a reduction in the use of photosynthetic products and consequently starch accumulation [24]. On the other hand, prolonged severe drought may cause mechanical damage, leading to metabolic disorders and blocked transport of stored NSC [7].

Fine root soluble sugars under severe drought treatment decreased between day 45 and 60, while needle and coarse root increased at day 60 (Figure 1). Our results are consistent with the fact that soluble sugar content in leaves and stems of gymnosperms increases under severe drought conditions [11]. Under drought conditions, gymnosperm soluble sugars in each organ of the tree increase to improve osmotic regulation to cope with the stress. Moreover, severe drought can impede the transfer of available carbon from leaves and stems to roots [13]. Therefore, the increase in stem starch under severe drought treatment at day 60 (Figure 2B) is also due to the accumulation of material as it is continuously obtained from photosynthesis but not transported out. And the decrease in soluble sugars of fine roots may also be due to the fact that fine roots are still growing but the transport pathway is blocked. As the degree of blockage is proportional to the intensity of drought, the material is unevenly replenished and consumed among organs [21].

The variation patterns in NSC of gymnosperms and angiosperms under drought stress are different [11]. There is still no consistent conclusion on the effect of drought stress on NSC, with some studies finding an increase in stored NSC in trees under drought stress [26,29], some finding an NSC decrease [30], and yet some others suggesting unchanged NSC [31]. This suggests that plants coordinate photosynthesis, growth and respiration through complex internal stabilization mechanisms to maintain the relative stability of NSC during drought. Further understanding of the inter-organ distribution of NSC in arid environments would require the use of other technologies such as carbon isotope tracers.

### 4.2. Effects of Drought Duration on NSC Dynamics

Research has demonstrated that timescales of various physiological processes in different drought stress stages are an important factor in understanding the NSC dynamics [32]. In the short term, tree growth is more sensitive to drought stress than photosynthesis, and may increase the accumulation of NSC [15,16]. This may be due to the role of stored carbon in the maintenance of various functions of the plant, sustaining homeostasis among and within the organs. During prolonged drought, storage NSC plays a role in maintaining the essential functions, such as osmoregulation and maintaining cellular tension [13], possibly also prolonging the time to death and maintaining xylem water potential [2].

In this study, needle, stem and coarse root NSC remained largely constant within the first 15 days, but fine root NSC decreased at the beginning of drought stress, whereas at day 60, stem NSC increased and fine root NSC decreased with increasing drought intensity. These results are consistent with our second hypothesis that NSC in *P. yunnanensis* does not change during the initial stages of drought stress, while it decreases in the later stages. This also suggests that trees may exhibit resistance stability in the early stages of drought, by allocating more carbon into storage organs to maintain their functionality [13]. In the medium term, the decline in photosynthesis causes a decrease in reserve NSC as they are used for metabolism [5,16]. Consistently, NSC contents in this study started to decrease at day 45 (Figure 3). In the long term, the complex adaptive mechanisms of trees may be fine-tuned between photosynthesis and growth to ensure sufficient stored NSC to maintain physiological regulation [33,34].

Soluble sugar content of needles and fine roots in severe and moderate drought treatments first increased and then decreased (Figure 1A–D) and starch content of needles in all treatments first decreased and then increased, while starch content of stems, coarse roots and fine roots remained stable in the first 15 days, and after that, it first increased and then decreased (Figure 2A–D). This may be due to the fact that when *P. yunnanensis* seedlings encounter drought stress, growth is first inhibited and soluble sugars of all organs increase in response to the organ’s need for osmoregulation [18,34]. Subsequently, the soluble sugar content of each organs decreased, probably due to the decrease in photosynthesis that reduced soluble sugars, and more importantly, prolonged drought increased the consumption of soluble sugars in coarse and fine roots. Furthermore, inter-organ transport was hindered and only locally stored NSC could be consumed.

In contrast, needle starch first decreased, which was probably converted to soluble sugars to maintain osmotic pressure. It then increased, probably because starch transport was blocked. Our results are similar to the NSC changes in *Robinia pseudoacaci* under drought stress [21]. Stem, coarse and fine root starch increased first, probably because of mobilization of stored carbon to maintain organ homeostasis; and later decreased, probably because they closed part of their stomata to maintain water balance [5], resulting in reduced photosynthesis, reduced sugar accumulation in leaves, and conversion of consumed starch to soluble sugars to maintain osmotic pressure.

In this study, during the 60-day period of drought treatment, the starch content of stems and fine roots of *Pinus yunnanensis* seedlings ranged from 20.47 mg·g^−1^ to 48.95 mg·g^−1^ and 9.10 mg·g^−1^ to 21.20 mg·g^−1^ (Figure 2A–D), respectively. The NSC content of stems and fine roots ranged from 77.14 mg·g^−1^ to 132.55 mg·g^−1^ and 43.08 mg·g^−1^ to 61.87 mg·g^−1^ (Figure 3A–D), respectively. This is consistent with previous studies that the NSC content did not approach zero [24]. It is probably because under drought stress, NSC content in plants has to maintain a certain threshold to sustain metabolic processes and hydrodynamic integrity [35]. Failure of osmoregulation or other factors will lead to tree death when NSC levels are reduced beyond a certain level [35]. Therefore, the mortality mechanism of *P. yunnanensis* seedlings under drought stress needs to be further investigated.

## 5. Conclusions

Our results show that drought duration had a significant effect on the content of nonstructural carbohydrates (NSCs) in all organs of *P. yunnanensis* seedlings, whereas drought intensity only had a significant effect on NSC in stems and fine roots. Within the first 45 days of drought stress, NSC contents in needle, stem and coarse foot remained largely unchanged, except for a decline in fine root NSC content under light drought treatment at day 15. However, by day 60, fine root NSC content decreased significantly under severe drought treatment, while stem NSC increased significantly under moderate and severe drought treatment. These findings provide great insights into the dynamic changes in NSC in different organs under drought stress.

However, this work has its limitations. Firstly, only changes in potted seedlings were discussed in this study. Whether such a pattern exists in seedlings and young trees under field conditions remains to be studied. In addition, this study did not control carbon input, nor lighting conditions when studying the effects of drought stress. How stored NSC varies in interaction with complete carbon limitation or circadian/light regulation are both promising directions for further research.

## Figures and Tables

**Figure 1 plants-14-00980-f001:**
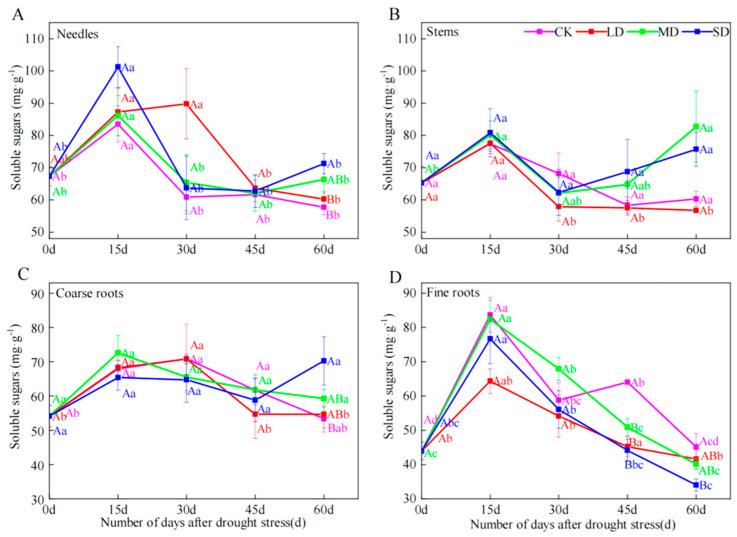
The effects of drought durations on the content of soluble sugar in needles (**A**), stems (**B**), coarse roots (**C**) and fine roots (**D**) of *P. yunnanensis* under four kinds of stresses: suitable moisture (CK), light drought (LD), moderate drought (MD) and severe drought (SD). Different lowercase letters indicate significant differences between different sampling dates; different capital letters indicate significant differences among different drought stress intensity (*p* < 0.05).

**Figure 2 plants-14-00980-f002:**
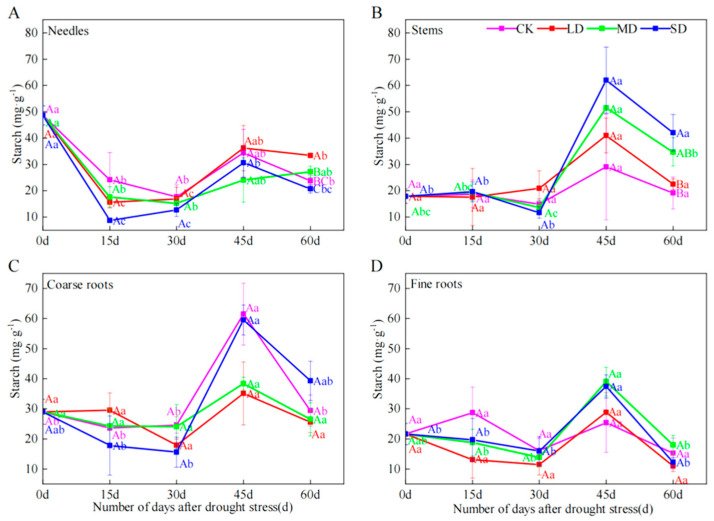
The effects of different durations on the content of starch in needles (**A**), stems (**B**), coarse roots (**C**) and fine roots (**D**) of *P. yunnanensis* under four kinds of stresses: suitable moisture (CK), light drought (LD), moderate drought (MD) and severe drought (SD). Different lowercase letters indicate significant differences between different sampling dates; different capital letters indicate significant differences among different drought stress intensity (*p* < 0.05).

**Figure 3 plants-14-00980-f003:**
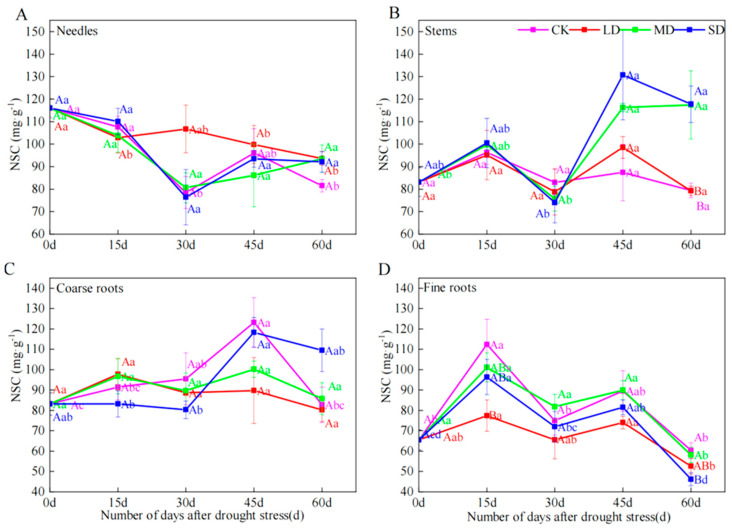
The effects of different durations on the content of NSC in needles (**A**), stems (**B**), coarse roots (**C**) and fine roots (**D**) of *P. yunnanensis* under four kinds of stresses: suitable moisture (CK), light drought (LD), moderate drought (MD) and severe drought (SD). Different lowercase letters indicate significant differences between different sampling dates; different capital letters indicate significant differences among different drought stress intensity (*p* < 0.05).

**Figure 4 plants-14-00980-f004:**
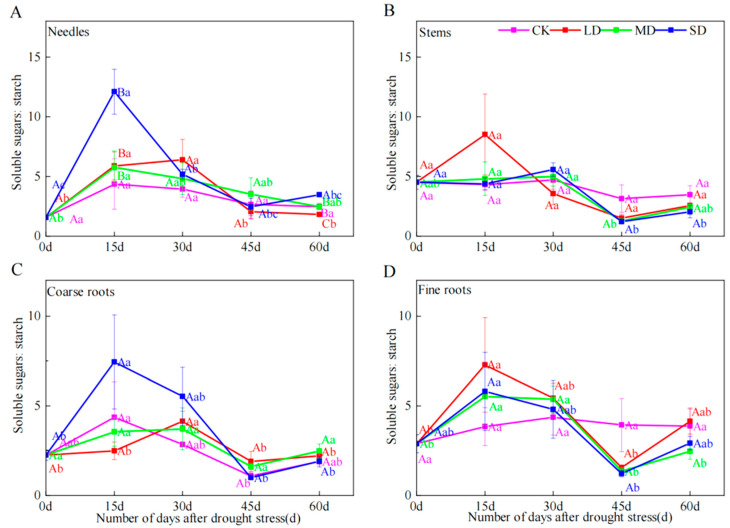
The effects of different durations on the ratio of soluble sugars to starch in needles (**A**), stems (**B**), coarse roots (**C**) and fine roots (**D**) of *P. yunnanensis* under four kinds of stresses: suitable moisture (CK), light drought (LD), moderate drought (MD) and severe drought (SD). Different lowercase letters indicate significant differences between different sampling dates; different capital letters indicate significant differences among different drought stress intensity (*p* < 0.05).

**Figure 5 plants-14-00980-f005:**
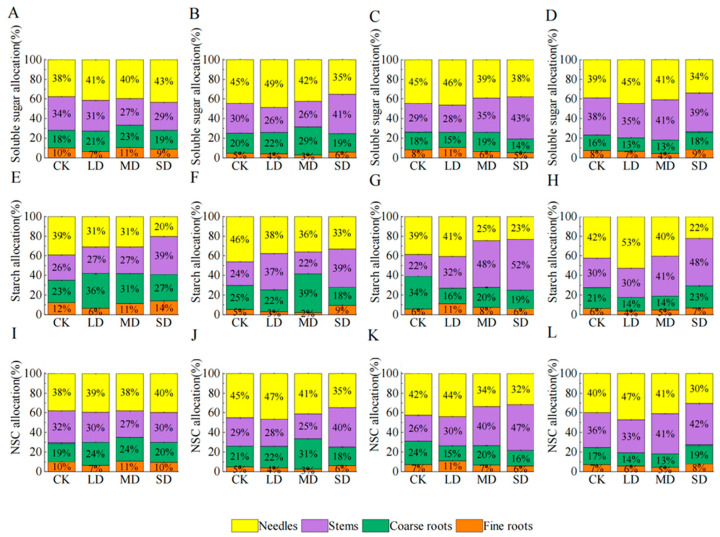
The effects of different drought stress durations ((**A**,**E**,**I**): 15 d; (**B**,**F**,**J**): 30 d; (**C**,**G**,**K**): 45 d; (**D**,**H**,**L**): 60 d) on the distribution patterns of NSC in needles, stems, coarse roots, fine roots, soluble sugars, starch, and NSC of *P. yunnanensis* under four kinds of stresses (suitable moisture (CK), light drought (LD), moderate drought (MD) and severe drought (SD)).

**Table 1 plants-14-00980-t001:** Two-way ANOVA (F-value) of different drought stress intensities, durations and organs on soluble sugar, starch, NSC and soluble sugar-to-starch ratios of *P. yunnanensis* seedlings. Note: * *p* < 0.05, ** *p* < 0.01.

	Fixed Factors	Soluble Sugar	Starch	NSC	Soluble Sugar: Starch
Needles	Drought intensity	2.10	2.04	2.46	3.66 *
	Drought duration	10.60 **	24.81 **	10.88 **	18.76 **
	Drought intensity × progression	1.67	0.60	0.99	2.69 **
Stems	Drought intensity	2.00	3.88 *	4.05 *	1.56
	Drought duration	4.08 **	16.80 **	5.45 **	9.98 **
	Drought intensity × progression	1.75	1.54	1.44	1.87
Coarse roots	Drought intensity	1.73	1.07	1.53	1.58
	Drought duration	6.26 **	9.45 **	4.31 **	6.76 **
	Drought intensity × progression	1.58	1.40	2.12 *	1.13
Fine roots	Drought intensity	5.77 **	1.95	5.83 **	0.55
	Drought duration	42.82 **	7.48 **	19.15 **	6.86 **
	Drought intensity × progression	1.76	1.16	0.69	1.35

## Data Availability

Data are contained within the article.

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
