# Peer review of "The Effects of Drought Stress Intensity and Duration on the Dynamics of Nonstructural Carbohydrates in Pinus yunnanensis Seedlings"

_plants, 2025, doi:10.3390/plants14060980_

Round 1
Reviewer 1 Report
Comments and Suggestions for Authors
The manuscript investigates the effect of drought stress intensity and duration on the dynamics of nonstructural carbohydrates (NSC) in P. yunnanensis seedlings. The study is relevant and timely, considering the increasing frequency of drought events due to climate change. The findings contribute to understanding the physiological adaptations of trees to drought stress, particularly in terms of carbon allocation. The methodology is generally sound, and the results are well presented. However, several issues require attention before the manuscript can be considered for publication.
- Consider making the title more precise, such as: "Effects of Drought Stress Intensity and Duration on Nonstructural Carbohydrate Dynamics in Pinus yunnanensis Seedlings."
- The abstract should highlight key findings more explicitly and mention any unexpected results.
- How were the seedlings selected to ensure uniformity in size and health before treatment?
- The conclusion that 60 d marks a "turning point" for NSC changes should be substantiated with more discussion on underlying physiological mechanisms.
- this paper discusses NSC changes in different organs but does not adequately explain how these changes impact plant survival and drought resilience.
- Were there any environmental fluctuations (e.g., temperature or humidity) during the 60-d period that could have influenced NSC accumulation?
- The discussion lacks coherence in linking the findings to broader ecological implications. Consider integrating more references to existing literature for better context.
- A thorough language revision is needed.
A thorough language revision is needed.
Author Response
- We have made revised the title.
- We have revised the abstract.
- The height and root collar diameter of the seedlings were measured to ensure uniformity in size. The health status of the seedlings was assessed through visual observation, combined with years of research experience on this particular tree species.
- The conclusion that 60 d marks a "turning point" for NSC changes. This conclusion is derived from the data, and the underlying physiological mechanisms have already been analyzed in the discussion from two perspectives (the duration of drought stress and the intensity of drought stress ) in different organs.
- Regarding how it affects plant survival, we only observed the plants for 60 days. A longer experimental period, extending until the plants die from drought, is needed to analyze the relationship between NSC and plant survival rate. In terms of drought resistance, this paper primarily focuses on the indicator of NSC, emphasizing the analysis of their dynamic changes. However, the assessment of drought resistance requires more indicators for a comprehensive evaluation.Currently, the patterns of NSC changes under drought stress are inconsistent. This study, using Pinus yunnanensis as the subject, focuses solely on elucidating the dynamics of NSC in various organs.
- This study focuses on the changes in non-structural carbohydrates, sugars, and starch in various organs at different times under drought stress. The research method employed is a pot experiment with precise water control, which is primarily a theoretical exploration and analysis. In terms of broader ecological implications, it is challenging for this study to expand from this perspective, as it requires the measurement of more indicators and relies on field observation experiments to draw conclusions.
- We specifically engaged individuals with strong language skills to revise and refine the language.
Reviewer 2 Report
Comments and Suggestions for Authors
Dear Authors,
I have read your manuscript with interest. The topic is significant and has the potential to contribute to the fuel research field; however, the manuscript requires careful revision. Please find my comments below:
1. The manuscript does not adhere to the correct format for MDPI journals (Plants). Please use the Plants template and ensure the manuscript follows the journal's formatting guidelines. Titles, subtitles, tables, and figures must align with the journal’s format. Additionally, the in-text citations and bibliography do not follow the required style. Please carefully review the journal’s author guidelines and make the necessary corrections.
2. The abstract should clearly reflect the differences between treatments based on quantitative data. Furthermore, please provide details on the treatment conditions rather than only mentioning treatment names.
3. The hypotheses appear to have been formulated after obtaining the results. Please revise them to ensure they are well-defined and based on the study's initial objectives.
4. Regarding the drought treatment: How was consistency maintained throughout the 60-day experimental period? Please clarify.
5. Please include a figure illustrating the different plant parts considered in this experiment (e.g., needles, stem, coarse root, fine roots).
6. Line 122: The in-text citation appears unnecessary, particularly since the cited study focuses on a different species. Additionally, citing a reference for moisture retention may not be required. Please reconsider its inclusion.
7. Line 158: Ensure consistency in terminology throughout the manuscript. For example, the terms coarse root and thick root should be unified.
8. Line 159: The phrase "128 fine roots" appears incorrect. Please revise for clarity.
9. Please check the entire manuscript for spelling and grammatical issues. For instance, in Line 159, "Wase equal" should be corrected.
10. Table 1: The formatting appears misaligned. Additionally, the table title refers to a two-way ANOVA, yet the data presented do not reflect interaction effects, suggesting that a one-way ANOVA may have been conducted instead. Please clarify. In the table description, provide all relevant details, including what the data represent, the number of replications, and any additional necessary information.
11. Some results are not well explained, such as Figure 4. Furthermore, it is important to justify the significance of each measured parameter. For instance, why was the soluble solids to starch ratio measured? Please clarify its relevance.
12. Figure 5: The figure does not include a statistical comparison between treatments; it only presents means with standard values. I suggest adding Tukey’s test letters or asterisks to indicate significant differences. Ideally, the test should be based on a two-way ANOVA. Additionally, it would be more effective to replace the stacked bar graph with a simple bar graph for clarity.
13. The Discussion should compare and contrast the study’s findings with previous research, highlighting both similarities and differences.
14. The Conclusion should be supported by the results and should logically follow from the Discussion section.
15. Please relocate the sections Acknowledgments, Author Contributions, Availability of Data and Materials, and Conflicts of Interest to appear before the References section.
Comments on the Quality of English LanguageThe English quality of the manuscript requires significant improvement, as there are numerous grammatical errors, awkward phrasings, and inconsistencies throughout the text. These issues affect the clarity and readability of the manuscript, making it difficult to follow certain arguments and interpretations. To enhance the overall quality of writing, I strongly recommend having the manuscript reviewed by a native English-speaking colleague or a professional language editing service. Ensuring clear and precise language will greatly improve the readability and impact of your research.
Author Response
- We have already revised the format accordingly.
- Modifications have been made in abstract.
- Our research hypothesis originates from a paper on the dynamic changes of NSC in Robinia pseudoacaciaunder drought stress. Based on the conclusions of that paper, we developed our study, and the hypothesis was formulated prior to obtaining the results, not afterward. [Zhang T, Cao Y, Chen Y M, Liu GB Non-structural carbohydrate dynamics in Robinia pseudoacacia saplings under three levels of continuous drought stress. Trees. 2015,29(6):1837-1849.]
- To minimize interference from other factors, we made every effort within our capabilities. During the entire experimental period, from March to May, the experiment was conducted in a greenhouse rather than in an open field. We strictly controlled the soil water content in accordance with technical standards, using the weighing method. Each day, we performed weighing and replenished water as needed. Specific details of this process have been outlined in the relevant section(Application of drought treatments).
- This experiment was conducted in 2020, and while some photos were taken at the time, the graduate students who were directly involved in the experiment no longer have these photos saved after several years, so they cannot be supplemented. The division of various organs was also carried out using methods commonly adopted by peers in the field.
- Here, we used the commonly adopted values for soil water content at different gradients in drought stress experiments within the field, specifically the percentage of field water holding capacity, as the reference for dividing the four soil moisture gradients. This information should be included here; otherwise, it might raise questions about why we set four gradients and the criteria for their division. Without this, the basis for defining the four gradients would appear unclear.
- Modifications have been made.
- Modifications have been made.
- Modifications have been made.
- We performed a two-way ANOVA in SPSS, not a one-way ANOVA, although the interaction effects were not significant. The data were based on 4-6 replicates, and the number of replicates is detailed in the Materials and Methods section.
- We only explained the changes in key data. In previous submissions, some experts pointed out that the results section was too lengthy and needed to be streamlined. Therefore, in the results description, we focused only on the main findings most relevant to our research topic. The ratio mentioned in Figure 4 is a commonly used indicator in studies on the dynamic changes of non-structural carbohydrates under drought stress. The ratio reflects the dynamic balance between the two datasets, as sugars and starch are interconvertible.
- This is an innovative aspect of our paper. Through the figure, we can visually observe the proportion of each organ, directly reflecting the allocation of non-structural carbohydrates. Here, it is simply a ratio, and there is no need to include significance markers, as the significance has already been explained in the preceding results figures. Additionally, the data calculation for this figure was highly labor-intensive, as it was derived by combining biomass data.
- In the discussion section, we have already addressed this. Specifically, we compared and analyzed our findings with the results of studies on the dynamic changes of NSC under drought stress in other species.
- Modifications have been made.
Round 2
Reviewer 1 Report
Comments and Suggestions for Authors
it can be accepted now
Author Response
Response to Reviewer 1 Comments
|
||
1. Summary |
|
|
I would like to express my sincere gratitude to the reviewers for their insightful comments. All feedback has been thoroughly addressed, and the corresponding revisions have been highlighted within the revised manuscript.
|
||
2. Point-by-point response to Comments and Suggestions for Authors |
||
Comments 1: [Consider making the title more precise, such as: "Effects of Drought Stress Intensity and Duration on Nonstructural Carbohydrate Dynamics in Pinus yunnanensis Seedlings."]
|
||
Response 1: [We have made revised the title.] Thank you for pointing this out. I/We agree with this comment. Therefore, I/we have revised the title[The Effects of Drought Stress Intensity and Duration on the Dynamics of Nonstructural Carbohydrates in Pinus yunnanensis Seedlings.] “[updated text in the manuscript if necessary]” |
||
Comments 2: [The abstract should highlight key findings more explicitly and mention any unexpected results.] |
||
Response 2: Agree. We have revised the abstract.] “[How drought impacts the allocation of nonstructural carbohydrates (NSC) in Pinus yunnanensis remains unclear. In this study, Pinus yunnanensis seedlings were subjected to four levels of drought stresses treatment during a 60 day experiment period suitable moisture (CK), light drought (LD), moderate drought (MD), and severe drought (SD). NSC in needles, stems, coarse roots, and fine roots were measured every two weeks. The distribution of NSC in P. yunnanensis seedlings varied with both drought stress intensity and duration, with different organ-specific patterns under increasing drought stress. Before the first 15 days, the intensity of drought stress had no significant effect on needle, stem and coarse root NSC contents, but decreased fine root NSC contents significantly. Between day 30 and 45, drought stress intensity showed no significant effect on NSC content in all organs. However, At 60 d, compared with CK, stems NSC concentrations under MD and SD increased by 47.92% and 48.23%, whereas fine root NSC concentrations decreased by 23.38% under SD conditions. With the extension of the drought duration, coarse root NSC increased while fine root NSC content decreased under SD conditions. Our results highlight the important role played by drought duration in controlling the NSC dynamics. Only fine root NSC decreased at the initial stage, and day 60 emerged as a turning point at which organ-level NSC changes became marked. These findings provide great insights to the understanding of organ-specific NSC dynamics under drought stress.]” |
||
Comments 3:[How were the seedlings selected to ensure uniformity in size and health before treatment?] |
||
Response 3: [The height and root collar diameter of the seedlings were measured to ensure uniformity in size. The health status of the seedlings was assessed through visual observation, combined with years of research experience on this particular tree species.] |
||
Comments 4:[The conclusion that 60 d marks a "turning point" for NSC changes should be substantiated with more discussion on underlying physiological mechanisms.] |
||
Response 4: [The conclusion that 60 d marks a "turning point" for NSC changes. This conclusion is derived from the data, and the underlying physiological mechanisms have already been analyzed in the discussion from two perspectives (the duration of drought stress and the intensity of drought stress ) in different organs.] |
||
Comments 5:[this paper discusses NSC changes in different organs but does not adequately explain how these changes impact plant survival and drought resilience.] |
||
Response 5: [Regarding how it affects plant survival, we only observed the plants for 60 days. A longer experimental period, extending until the plants die from drought, is needed to analyze the relationship between NSC and plant survival rate. In terms of drought resistance, this paper primarily focuses on the indicator of NSC, emphasizing the analysis of their dynamic changes. However, the assessment of drought resistance requires more indicators for a comprehensive evaluation.Currently, the patterns of NSC changes under drought stress are inconsistent. This study, using Pinus yunnanensis as the subject, focuses solely on elucidating the dynamics of NSC in various organs.] |
||
Comments 6:[Were there any environmental fluctuations (e.g., temperature or humidity) during the 60-d period that could have influenced NSC accumulation?] |
||
Response 6: The experiment started on March 14 and ended on May 12, 2021, lasting 60 days in total. The experiment was conducted in a test greenhouse, where temperature and humidity remained relatively stable during the trial period. |
||
Comments 7:[The discussion lacks coherence in linking the findings to broader ecological implications. Consider integrating more references to existing literature for better context.] |
||
Response 7: [This study focuses on the changes in non-structural carbohydrates, sugars, and starch in various organs at different times under drought stress. The research method employed is a pot experiment with precise water control, which is primarily a theoretical exploration and analysis. In terms of broader ecological implications, it is challenging for this study to expand from this perspective, as it requires the measurement of more indicators and relies on field observation experiments to draw conclusions.] |
||
Comments 8:[A thorough language revision is needed.] |
||
Response 8: [We specifically engaged individuals with strong language skills to revise and refine the language.] |

Reviewer 2 Report
Comments and Suggestions for Authors
Dear Authors,
Thank you for submitting the revised version of your manuscript. I have thoroughly re-evaluated the revisions and found that only comments 1, 2, 7, 8, 9, and 14 have been addressed. However, 8 other comments remain unresolved. For your reference, I am providing my previous comments again. Please review them carefully and revise the manuscript accordingly. While I understand the situation regarding plant parts, I find the responses to the other comments insufficient. The goal of peer review is to enhance the quality of the manuscript. I kindly request that you make the necessary revisions and provide scientifically sound responses.
Please reconsider the comments other than 1,2,7,8,9,14.
- We have already revised the format accordingly.
- Modifications have been made in abstract.
- Our research hypothesis originates from a paper on the dynamic changes of NSC in Robinia pseudoacaciaunder drought stress. Based on the conclusions of that paper, we developed our study, and the hypothesis was formulated prior to obtaining the results, not afterward. [Zhang T, Cao Y, Chen Y M, Liu GB Non-structural carbohydrate dynamics in Robinia pseudoacacia saplings under three levels of continuous drought stress. Trees. 2015,29(6):1837-1849.]
- To minimize interference from other factors, we made every effort within our capabilities. During the entire experimental period, from March to May, the experiment was conducted in a greenhouse rather than in an open field. We strictly controlled the soil water content in accordance with technical standards, using the weighing method. Each day, we performed weighing and replenished water as needed. Specific details of this process have been outlined in the relevant section(Application of drought treatments).
- This experiment was conducted in 2020, and while some photos were taken at the time, the graduate students who were directly involved in the experiment no longer have these photos saved after several years, so they cannot be supplemented. The division of various organs was also carried out using methods commonly adopted by peers in the field.
- Here, we used the commonly adopted values for soil water content at different gradients in drought stress experiments within the field, specifically the percentage of field water holding capacity, as the reference for dividing the four soil moisture gradients. This information should be included here; otherwise, it might raise questions about why we set four gradients and the criteria for their division. Without this, the basis for defining the four gradients would appear unclear.
- Modifications have been made.
- Modifications have been made.
- Modifications have been made.
- We performed a two-way ANOVA in SPSS, not a one-way ANOVA, although the interaction effects were not significant. The data were based on 4-6 replicates, and the number of replicates is detailed in the Materials and Methods section.
- We only explained the changes in key data. In previous submissions, some experts pointed out that the results section was too lengthy and needed to be streamlined. Therefore, in the results description, we focused only on the main findings most relevant to our research topic. The ratio mentioned in Figure 4 is a commonly used indicator in studies on the dynamic changes of non-structural carbohydrates under drought stress. The ratio reflects the dynamic balance between the two datasets, as sugars and starch are interconvertible.
- This is an innovative aspect of our paper. Through the figure, we can visually observe the proportion of each organ, directly reflecting the allocation of non-structural carbohydrates. Here, it is simply a ratio, and there is no need to include significance markers, as the significance has already been explained in the preceding results figures. Additionally, the data calculation for this figure was highly labor-intensive, as it was derived by combining biomass data.
- In the discussion section, we have already addressed this. Specifically, we compared and analyzed our findings with the results of studies on the dynamic changes of NSC under drought stress in other species.
- Modifications have been made.
The English quality of the manuscript still needs to be improved.
Author Response
Response to Reviewer 2 Comments
|
||
1. Summary |
|
|
I would like to express my sincere gratitude to the reviewers for their insightful comments. All feedback has been thoroughly addressed, and the corresponding revisions have been highlighted within the revised manuscript.
|
||
2. Point-by-point response to Comments and Suggestions for Authors |
||
Comments 1: [The manuscript does not adhere to the correct format for MDPI journals (Plants). Please use the Plants template and ensure the manuscript follows the journal's formatting guidelines. Titles, subtitles, tables, and figures must align with the journal’s format. Additionally, the in-text citations and bibliography do not follow the required style. Please carefully review the journal’s author guidelines and make the necessary corrections.] |
||
Response 1: [We have already revised the format accordingly.] |
||
Comments 2: [The abstract should clearly reflect the differences between treatments based on quantitative data. Furthermore, please provide details on the treatment conditions rather than only mentioning treatment names.] |
||
Response 2: [Modifications have been made in abstract.] |
||
Comments 3:[The hypotheses appear to have been formulated after obtaining the results. Please revise them to ensure they are well-defined and based on the study's initial objectives.] |
||
Response 3: [Our research hypothesis originates from a paper on the dynamic changes of NSC in Robinia pseudoacaciaunder drought stress. Based on the conclusions of that paper, we developed our study, and the hypothesis was formulated prior to obtaining the results, not afterward. [Zhang T, Cao Y, Chen Y M, Liu GB Non-structural carbohydrate dynamics in Robinia pseudoacacia saplings under three levels of continuous drought stress. Trees. 2015,29(6):1837-1849.]] |
||
Comments 4:[Regarding the drought treatment: How was consistency maintained throughout the 60-day experimental period? Please clarify.] |
||
Response 4: [4.To minimize interference from other factors, we made every effort within our capabilities. During the entire experimental period, from March to May, the experiment was conducted in a greenhouse rather than in an open field. We strictly controlled the soil water content in accordance with technical standards, using the weighing method. Each day, we performed weighing and replenished water as needed. Specific details of this process have been outlined in the relevant section(Application of drought treatments).] |
||
Comments 5:[Please include a figure illustrating the different plant parts considered in this experiment (e.g., needles, stem, coarse root, fine roots).] |
||
Response 5: [5.This experiment was conducted in 2021, and while some photos were taken at the time, the graduate students who were directly involved in the experiment no longer have these photos saved after several years, so they cannot be supplemented. The division of various organs was also carried out using methods commonly adopted by peers in the field.] |
||
Comments 6:[Line 122: The in-text citation appears unnecessary, particularly since the cited study focuses on a different species. Additionally, citing a reference for moisture retention may not be required. Please reconsider its inclusion.] |
||
Response 6: Here, we used the commonly adopted values for soil water content at different gradients in drought stress experiments within the field, specifically the percentage of field water holding capacity, as the reference for dividing the four soil moisture gradients. This information should be included here; otherwise, it might raise questions about why we set four gradients and the criteria for their division. Without this, the basis for defining the four gradients would appear unclear. |
||
Comments 7:[Line 158: Ensure consistency in terminology throughout the manuscript. For example, the terms coarse root and thick root should be unified.] |
||
Response 7: [odifications have been made.] |
||
Comments 8:[Line 159: The phrase "128 fine roots" appears incorrect. Please revise for clarity.] |
||
Response 8:[Modifications have been made.] |
||
Comments 9:[Please check the entire manuscript for spelling and grammatical issues. For instance, in Line 159, "Wase equal" should be corrected.] |
||
Response 9: [Modifications have been made.] |
||
Comments 10:[ Table 1: The formatting appears misaligned. Additionally, the table title refers to a two-way ANOVA, yet the data presented do not reflect interaction effects, suggesting that a one-way ANOVA may have been conducted instead. Please clarify. In the table description, provide all relevant details, including what the data represent, the number of replications, and any additional necessary information.] |
||
Response 10: [We performed a two-way ANOVA in SPSS, not a one-way ANOVA, although the interaction effects were not significant. The data were based on 4-6 replicates, and the number of replicates is detailed in the Materials and Methods section.] |
||
Comments 11:[ Some results are not well explained, such as Figure 4. Furthermore, it is important to justify the significance of each measured parameter. For instance, why was the soluble solids to starch ratio measured? Please clarify its relevance.] |
||
Response 11: [We only explained the changes in key data. In previous submissions, some experts pointed out that the results section was too lengthy and needed to be streamlined. Therefore, in the results description, we focused only on the main findings most relevant to our research topic. The ratio mentioned in Figure 4 is a commonly used indicator in studies on the dynamic changes of non-structural carbohydrates under drought stress. The ratio reflects the dynamic balance between the two datasets, as sugars and starch are interconvertible.] |
||
Comments 12:[ Figure 5: The figure does not include a statistical comparison between treatments; it only presents means with standard values. I suggest adding Tukey’s test letters or asterisks to indicate significant differences. Ideally, the test should be based on a two-way ANOVA. Additionally, it would be more effective to replace the stacked bar graph with a simple bar graph for clarity.] |
||
Response 12: [This is an innovative aspect of our paper. Through the figure, we can visually observe the proportion of each organ, directly reflecting the allocation of non-structural carbohydrates. Here, it is simply a ratio, and there is no need to include significance markers, as the significance has already been explained in the preceding results figures. Additionally, the data calculation for this figure was highly labor-intensive, as it was derived by combining biomass data.] |
||
Comments 13:[ The Discussion should compare and contrast the study’s findings with previous research, highlighting both similarities and differences.] |
||
Response 13: [In the discussion section, we have already addressed this. Specifically, we compared and analyzed our findings with the results of studies on the dynamic changes of NSC under drought stress in other species.] |
||
Comments 14:[ The Conclusion should be supported by the results and should logically follow from the Discussion section.] |
||
Response 14: [Modifications have been made.] |
||
Comments 15:[ Please relocate the sections Acknowledgments, Author Contributions, Availability of Data and Materials, and Conflicts of Interest to appear before the References section.] |
||
Response 15: [Modifications have been made.] |

Round 3
Reviewer 2 Report
Comments and Suggestions for Authors
Dear Authors,
In my second review, I requested revisions based on the first round of comments; however, I could not identify any substantial improvements in the manuscript. Instead, it appears that you have only pasted your previous responses without making the necessary changes. Additionally, the manuscript has not been formatted according to the Plants journal template.
I kindly urge you to carefully revise the manuscript by incorporating the suggested improvements. Please ensure that all comments are thoroughly addressed and that the manuscript adheres to the required journal format. Below are my specific recommendations for revision:
- Line 122: The in-text citation appears unnecessary, particularly since the cited study focuses on a different species. Additionally, citing a reference for moisture retention may not be required. Please reconsider its inclusion.
- Some results are not well explained, such as Figure 4. Furthermore, it is important to justify the significance of each measured parameter. For instance, why was the soluble solids to starch ratio measured? Please clarify its relevance.
- Figure 5: The figure does not include a statistical comparison between treatments; it only presents means with standard values. I suggest adding Tukey’s test letters or asterisks to indicate significant differences. Ideally, the test should be based on a two-way ANOVA. Additionally, it would be more effective to replace the stacked bar graph with a simple bar graph for clarity.
Author Response
We sincerely appreciate the valuable feedback provided by the reviewers. Our team has carefully reviewed all comments and implemented the suggested revisions in the revised manuscript, with detailed annotations highlighting the changes.
We confirm that all reviewer concerns have been thoroughly addressed. Should further refinements be needed, we are fully committed to incorporating any additional feedback.Thank you for your guidance in strengthening this work.
Comments 1: Line 122: The in-text citation appears unnecessary, particularly since the cited study focuses on a different species. Additionally, citing a reference for moisture retention may not be required. Please reconsider its inclusion.
Response 1: Modifications have been made. We have removed the citation in question, as suggested.
Comments 2: Some results are not well explained, such as Figure 4. Furthermore, it is important to justify the significance of each measured parameter. For instance, why was the soluble solids to starch ratio measured? Please clarify its relevance.
Response 2: Modifications have been made. We have explained in the text and supplemented the results.
Comments 3: Figure 5: The figure does not include a statistical comparison between treatments; it only presents means with standard values. I suggest adding Tukey’s test letters or asterisks to indicate significant differences. Ideally, the test should be based on a two-way ANOVA. Additionally, it would be more effective to replace the stacked bar graph with a simple bar graph for clarity.
Response 3:We appreciate the reviewer’s insightful suggestion. To clarify, Figure 5 was designed to visualize the proportional allocation of sugars and starch across different plant organs under drought stress (expressed as percentages). The inclusion of numerical labels within the stacked bars directly highlights these proportions, akin to how survival rates are typically presented in ecological studies. This approach prioritizes clarity in showing how resources are dynamically partitioned among organs over time, rather than emphasizing statistical differences between treatments.
Our team deliberated extensively on whether to perform ANOVA/Tukey comparisons for this figure. However, we concluded that statistical testing of proportional data (e.g., percentage allocations) could introduce interpretational complexities, as the focus here is on relative distribution patterns rather than absolute quantitative differences. This methodology aligns with prior studies in the journal of Tree Physiology, where proportional allocations were analyzed without statistical annotations to emphasize resource partitioning trends.
Round 4
Reviewer 2 Report
Comments and Suggestions for Authors
Dear Authors,
Thank you for your efforts in revising the manuscript.
Regarding Figure 5, I recommend changing it to a bar graph for better visualization. Since you have included error bars, it suggests that you have replicates. Therefore, performing Tukey’s test would allow for a more robust comparison of treatments.
Author Response
Dear reviewer, I have modified Figure 5.We adopt the conventional expression method of this type of data in academic papers, remove the error line, and more obviously see the distribution changes of NSC in each organ at each stage.